

# Development of a new distributed hybrid seismic-electrical data acquisition station based on system–on-a-programmable-chip technology

Qisheng Zhang[1], Wenhao Li[1*], Feng Guo[1], Zhenzhong Yuan[1], Shuaiqing Qiao[1], Qimao Zhang[2]

[1]School of Geophysics and Information Technology, China University of Geosciences (Beijing), Beijing, China

[2]Institute of Electronics, Chinese Academy of Sciences, Beijing, China

**Correspondence:** Wenhao Li (li.wh@cugb.edu.cn)

**Abstract.** In the past few decades, with the continuous advancement of technology, seismic-electrical instruments have developed rapidly. However, complex and harsh exploration environments have put forward higher requirements and severe challenges for traditional geophysical exploration methods and instruments. Therefore, it is extremely urgent to develop new high-precision exploration instruments and data acquisition systems. In this study, a new distributed seismic-electrical hybrid acquisition station is developed using system-on-a-programmable-chip (SoPC) technology. The acquisition station hardware includes an analog board and a main control board. The analog board uses a signal conditioning circuit and a 24-bit analog-to-digital converter (ADS1271) to achieve high-precision data acquisition, while the main control board uses a low-power SoPC chip to enable high-speed stable data transmission. Moreover, the data transmission protocol for the acquisition station was designed, an improved low-voltage differential signaling data transmission technology was independently developed, and a method to enhance the precision of synchronous acquisition was studied in depth. These key technologies, which were developed for the acquisition station, were integrated into the SoPC of the main control board. Testing results indicate that the synchronization precision of the acquisition station is better than 200 ns, and the maximum low-power data transmission speed is 16 Mbps along a 55 m cable. Simultaneously, the developed acquisition station has the advantages of low noise, large dynamic range, low power consumption, etc., and it can achieve high-precision hybrid acquisition of seismic-electrical data.

## 1 Introduction

Geophysics is a discipline in which physics principles, methods, and instruments are applied to research and our understanding of the Earth and near-Earth space. It therefore plays an important role in the exploration and development of energy and mineral resources (Liu, 2017). The advancement of earth science is inseparable from high-end geoscience instruments (Liu, 2013). As early as the 1970s, the French company SERCEL had already introduced the SN338 digital seismograph and its single-station single-channel acquisition station (Huang and Yu, 1994). Presently, the most advanced seismic data acquisition systems in the world include SERCEL's 508XT system, INOVA's G3i HD system, and FireFly's DR31 system. The 508XT system uses intelligent network technology to achieve autonomous acquisition and local storage, as well as real-time quality control and



data transmission to the central recording unit (Van et al., 2001; Mrmureanu et al., 2011). In 2018, Dynamic Technologies (DTCC) Group launched the industry's first intelligent seismic sensor, SmartSolo, that integrates high-sensitivity geophones with acquisition circuits, and thus possesses advantages, such as compact size, light weight, and low cost. In terms of electrical exploration, developed countries not only started early in the theoretical research of electrical exploration technology, but also

made major breakthroughs in research and development as well as industrialization of electrical exploration instruments. Representative electrical exploration instruments include the GEODE EM3D system from US company GEOMETRIC for the controlled source audio-frequency magnetotellurics (CSAMT)method, the KMS-820 system from US company KMS, the GDP series multi-function electrical instrument from US company Zonge, the 8th generation system V8 from Canadian company Phoenix, the distributed transient electromagnetic system GEOFERRET from Australian company WMC, and the

ADU-08E system from German company METRONIX (Chen et al., 2018). It can be inferred that foreign seismic-electrical exploration instruments have reached a high level in terms of performance, stability, and industrialization, and they are being developed to achieve intelligence and multi-functionality.

For any country in the world, excessive reliance on foreign oil and gas resources can be seen as a safety warning. Yet, China was the world's largest importer of crude oil and natural gas in 2018 (CNPC Research Institute of Economics and Technology,

2019), which poses a serious threat to economic and energy security. Therefore, China must strengthen its exploration and development of domestic oil and gas resources to provide a strong guarantee for the sustained, stable, and healthy development of the domestic economy. However, China's oil and gas resources exploration has now entered a difficult stage, as increasingly complex geological areas and severe exploration environments have brought new requirements and severe challenges to traditional geophysical exploration methods and instruments. It is hence imminently required to adopt new exploration methods

and develop new exploration instruments that are suitable for China's national conditions.

In recent years, research and development of seismic and electrical instruments in China has achieved certain results as the state puts more emphasis on these areas. The Eastern Geophysical Exploration Co., Ltd. of China National Petroleum Corporation (CNPC) has developed a new generation of Chinese-made seismic exploration node instruments, eSeis1.0. The Institute of Geology and Geophysics of the Chinese Academy of Sciences has designed, based on computer network, a digital

seismic acquisition system with low power consumption that is capable of data transmission through hundreds of thousands of channels. Jilin University developed a wired telemetry seismograph, GEIST 438, using an Ethernet relay solution, while the University of Science and Technology of China adopted long-distance Ethernet physical layer technology to achieve a peak transmission rate of 40 Mbps (Qiao et al., 2018; Zhang et al., 2017; Guo et al., 2012; Zhang, 2007). Simultaneously, many Chinese institutes have also carried out research and development of electrical instruments. The Institute of Geophysical and

Geochemical Exploration of the Chinese Academy of Geological Sciences began to study a distributed passive-source electromagnetic (DPEM) system in 1991 and have tested its application many times in the exploration of mineral and groundwater resources. Relevant technical indicators of the Surface Electromagnetic Prospecting (SEP) system completed in collaboration by the Institute of Geology and Geophysics and the Institute of Electronics of the Chinese Academy of Sciences have reached the international standard. In the field of transient electromagnetic (TEM) research, Jilin University has



developed the ATEM-II system, while the Institute of Geophysical and Geochemical Exploration of the Chinese Academy of Geological Sciences has developed the IGGETEM and IGGETEM-20 systems (Di et al., 2013; Lin et al., 2014; Chen et al., 2010; Cheng et al., 2019). Yet, from a general point of view, China has not fully mastered the core technology in the development of geophysical instruments, and there is still a big gap compared with international leaders. Simultaneously, a

problem has arisen from the fact that a single geophysical exploration method has multiple solutions, and this has been bothering geophysicists for years. To make the best use of various exploration methods, so as to reduce the possibility of multiple solutions and improve the reliability of exploration results, research on the joint inversion of seismic-electrical methods from within and outside of China was started many years ago. Nevertheless, research on integrated seismic-electrical hybrid acquisition technology is still in its infancy. To achieve high-resolution hybrid seismic-electrical data acquisition with

high-precision synchronization and high-speed data transmission under low-power conditions is still a challenge for researchers and developers to overcome. Therefore, the development of a new seismic-electrical hybrid acquisition station, for which the intellectual property rights are exclusively held, is of great significance for improving the level of geophysical instruments and equipment and promoting the development of combined seismic-electrical exploration technology in China.

## 2 System-on-a-programmable-chip technology and distributed telemetry data acquisition system

### 2.1 System-on-a-programmable-chip technology

US company Altera Corporation (which was acquired by Intel Corporation) first introduced system-on-a-programmable chip (SoPC) technology in 2000. A SoPC is a flexible and highly efficient system-on-a-chip (SoC) solution, which integrates functional modules required by the system, such as a processor, I/O interface, hardware accelerator, memory, etc., into a field programmable gate array (FPGA) device to form a programmable SoC (Astarloa et al., 2005). A SoPC provides developers

with a flexible design approach with several advantages, such as low power consumption, easy upgrades, and modifications, and both its hardware and software are programmable (Zhang et al., 2012). With continuous improvement of the performance of programmable logic devices and the development of electronic design automation technology, SoPC technology has been applied in many fields. In this study, SoPC technology is used to design a new hybrid seismic-electrical acquisition station. The advantages of SoPC technology improve the overall performance of the acquisition station. Key technologies of the

distributed telemetry seismic-electrical hybrid acquisition station based on SoPC technology will be highlighted.

### 2.2 Components of the distributed telemetry data acquisition system

The distributed telemetry data acquisition system mainly consists of sensors, acquisition stations, power stations, interconnection stations, a central station, and communication cables. Figure 1 shows a block diagram of the system, in which sensors and acquisition stations form the front-end acquisition device and acquisition stations and power stations are connected

by cables, while interconnection stations and the central station, as well as the interconnection stations themselves, are connected by optical fibers. Power stations and interconnection stations power the acquisition stations through Power over



Ethernet (PoE) and simultaneously carry out acquisition data transfer. The central station monitors the operation of the entire acquisition system and performs data recovery (Wang, 2010).

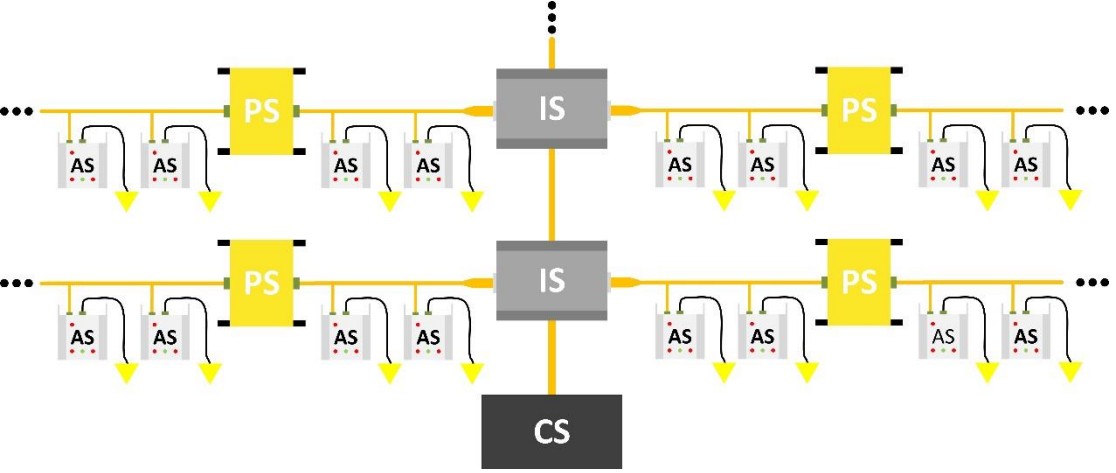

**Figure 1: Block diagram of the distributed telemetry data acquisition system. AS, acquisition station; PS, power station; IS, interconnection station; CS, central station.**

## 2.3 Overall framework of the new distributed hybrid seismic-electrical data acquisition station based on SoPC technology

The acquisition station is the key unit for obtaining seismic-electrical data. Figure 2 shows the overall architecture of the acquisition station developed in this study, which adopts ADS1271+SoPC to achieve high-precision analog-to-digital conversion and high-speed real-time data transmission. ADS1271 is a 24-bit Δ-Σ analog-to-digital converter (ADC) with excellent DC precision and AC performance. Analog signal output by sensors (detectors or electrodes) is passed through signal conditioning circuits for filtering, amplification, and so on, to then be input into the ADS1271 for analog-to-digital conversion to generate a digital signal. The data can be further filtered and synchronized with high precision, and it is then transmitted to the corresponding power station using Manchester encoding and custom data transmission protocols, as well as the independently developed low-voltage differential signaling (LVDS) low-power data transmission technology. The acquisition station developed in this study has the following characteristics: (1) A single station integrates high-precision acquisition channels of seismic and electrical data, which can be used for integrated seismic-electrical exploration; (2) Key technologies of the developed acquisition station are integrated in a single SoPC, the degree of integration is high, and intellectual property rights are exclusively held.





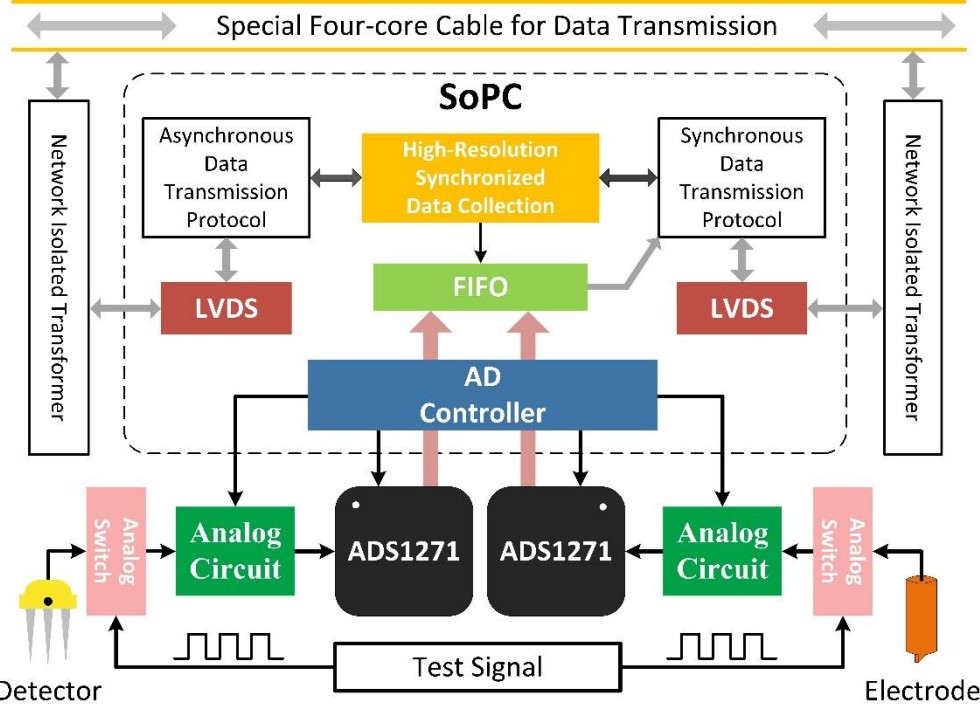

Figure 2: The overall structure of the data-acquisition station.

## 3 Key technologies of the new distributed hybrid seismic-electrical data acquisition station

### 3.1 Hardware circuit design of the data acquisition station

The hardware circuit of the acquisition station is composed of an analog board and a main control board. The analog board is mainly used for signal acquisition, conditioning, and high-precision analog-to-digital conversion. The main control board achieves the control of each unit in the analog board as well as data encoding and transmission.

Figure 3 shows a block diagram of the analog board, which has two data acquisition channels for simultaneous acquisition of seismic and electrical data. The analog board uses the AD780 ultrahigh precision reference voltage source to generate a 2.5 V

calibration signal, which is used to perform a self-test on the acquisition channels. An ADG419 chip is used as the analog switch that selects either the detector signal or the calibration signal to enter the low-pass filter circuit, after which the programmable-gain instrumentation amplifier, AD8253, performs program-controlled amplification of the signal. The signal then enters a drive circuit, THS4521, to be converted into a differential signal that finally undergoes high-precision analog-to-digital conversion in ADS1271. Design of the electrical channel is similar to that of the seismic channel. The difference is that

an LT1168 precision instrumentation amplifier is used to pre-amplify the signal in the amplifier circuit. Simultaneously, to reduce the interference due to the 50 Hz utility frequency, four operational amplifiers, OPA4227, are used to build the filter circuit, as shown in Fig. 4. The principle behind this is to construct a band-pass filter by using three operational amplifiers and



subtract the signal through the band-pass filter from that of the all-pass circuit using the fourth operational amplifier to achieve filtering of the 50 Hz utility frequency signal.

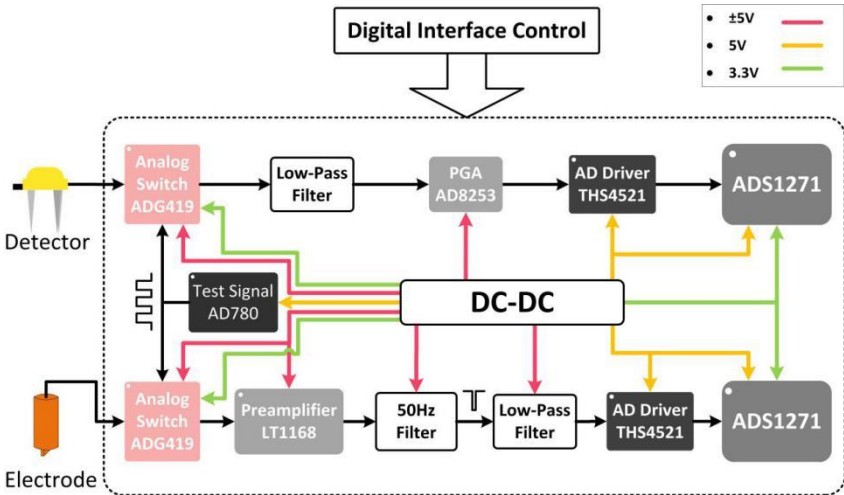

**Figure 3: Block diagram of the analog board of the data-acquisition station.**

5 The voltage of the power station continuously reduces as it supplies power to multiple acquisition stations in a PoE-like manner, and as a result, the voltage entering the acquisition stations varies from 22 to 48 V (determined by the position of each acquisition station). Since the voltages required for the analog board are ±5 V and 3.3 V, a PWB4812MD DC-DC chip is used to first convert the input voltage to 12 V, then a TMR3-1221WI conversion chip and PTH08080W switched-mode power supply are used to convert 12 V to ±5 V and 3.3 V, respectively.

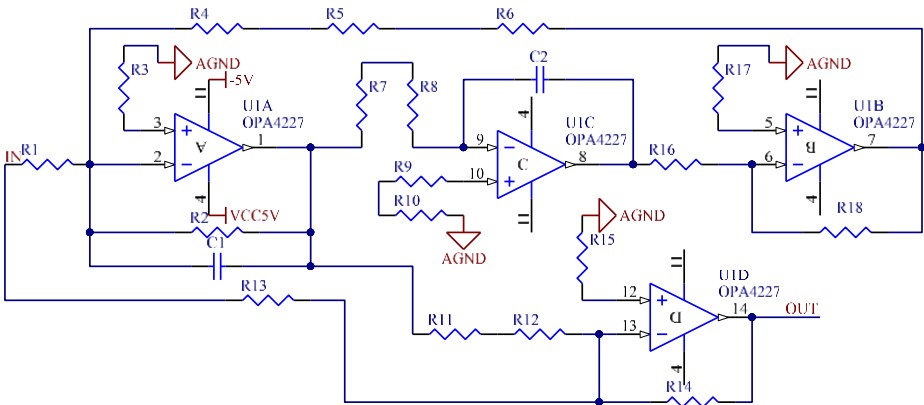

**Figure 4: 50 Hz filter circuit schematic.**

The main control board of the acquisition station is primarily composed of the SoPC chip, 5CEBA2F17I7N, and its configuration chip, EPCS16. Figure 5 shows a block diagram of the main control data transmission board. The 5CEBA2F17I7N chip is a Cyclone V series FPGA, whose total power consumption compared with the previous generation has dropped by 40% due to the 28 nm low-power process technology. The main control board uses four pairs of LVDS



differential pins on the SoPC and two network isolation transformers, MS10232NL, for data transmission. The advantages of using differential signaling include low power consumption, low bit error rate, low crosstalk, and low noise. (R2, T1) and (R4, T3) are two pairs of transceivers, while V1 and V2 are two lines with a common-mode voltage that are trapped between (R2, T3) and (T1, R4), respectively. The difference in value between the two is recognized by the bridge rectifier circuit, R, as a

positive or negative voltage, and then transmitted to the analog board for voltage conversion. The digital and power interfaces are used for communication and power supply between the main control board and the analog board.

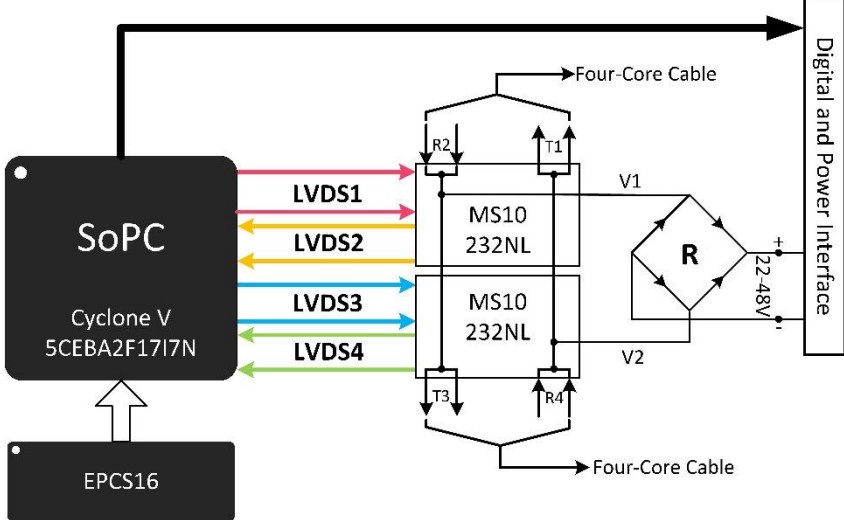

**Figure 5: Block diagram of the main control data transmission board.**

**3.2 Research on the dedicated data-transmission protocol of the data acquisition station**

An isolation transformer is introduced during LVDS data transmission, which will block the DC component of the signal. As a result, when 0 or 1 appears continuously in the data, the transformer will cause the voltage at the receiving end to drop, leading to the generation of jitters and bit errors in the signal. To solve this problem, this study uses the Manchester encoding technique which represents 0 and 1 based on voltage transitions. Manchester encoding makes it easier to extract a synchronized clock from the signal, and at the receiving end it constantly aligns the valid edges using a method for oversampling. This can

reduce the hardware circuit design, as well as the system power consumption, and simultaneously avoid the problem of burrs along the data transition edges.

The encoding module at the transmitting end first converts the 8-bit parallel data in units of bytes into a bit stream output by bit, and then transmits the encoded serial data in which "01" is used to represent the binary "0" and "10" is used to represent the binary "1". The data transmission rate on the transmission line is 16.384 Mbps during the process, meaning that the required

encoding clock should be 32.768 MHz. At the receiving end, to detect effective transitions in the Manchester encoded data stream and extract the synchronous clock, a method for oversampling is adopted. The sampling clock frequency used by the decoding module is 8 times that of the data transmission rate, and data sampling, decoding, and shift and byte synchronization





are carried out. Through this method of encoding/decoding, serial binary data transmission is achieved in the true sense that is enabled only by solving the problems of byte synchronization and frame synchronization. When the receiving end starts to receive one frame of data after receiving the preamble, it first receives the frame header that contains information on the data length of this frame, it then receives data according to the length defined in the frame header, and finally restores this data to

parallel data (in bytes).

Based on the characteristics of seismic-electrical data acquisition, a custom data frame was designed to be divided into 51 units, consisting of 816 bytes in total. The first unit is the frame header, and the remaining 50 units are data units for communication between power stations and acquisition stations, as well as between power stations themselves. The data unit follows a specific format, as shown in Table 1, which is arranged in an order from low byte to high byte, starting with the unit

header, command, status information, sampled data, and cyclic redundancy check (CRC), and it consists of 16 bytes in total. Within the data unit, sampled data occupies 12 bytes (or 4 data in total) and the remaining parts each occupy 1 byte.

**Table 1: Cell format of data frames in an acquisition station.**

| Byte 1 | Byte 2 | Byte 3 | Byte 4~6 | Byte 7~9 | Byte 10~12 | Byte 13~15 | Byte 16 |
|---|---|---|---|---|---|---|---|
| Head | Command | Status | Data1 | Data2 | Data3 | Data4 | CRC |

Data in the acquisition station can be transmitted in two modes: synchronous transmission and asynchronous transmission. As

shown in Fig. 6, the asynchronous transmission mode is adopted when reporting status information and uploading data from the slave power station to the master power station; while synchronous transmission mode is adopted when uploading data and status information from the acquisition station to the slave power station or the master power station, as well as when receiving control commands from the master power station (Li, 2018). During synchronous acquisition, the master power station sends out empty data frames. Each acquisition station then decodes these empty data frames, traverses them to its corresponding unit,

writes the data in the First Input First Output (FIFO)to the corresponding position of this unit, then sets the unit status byte to 1 and command byte to 0x0F, and finally recalculates the CRC for this unit. In asynchronous transmission, the received data only needs to be re-encoded and sent out without modifying the data frame. Therefore, acquisition stations in asynchronous transmission only undertake the task of data transfer between the power stations.

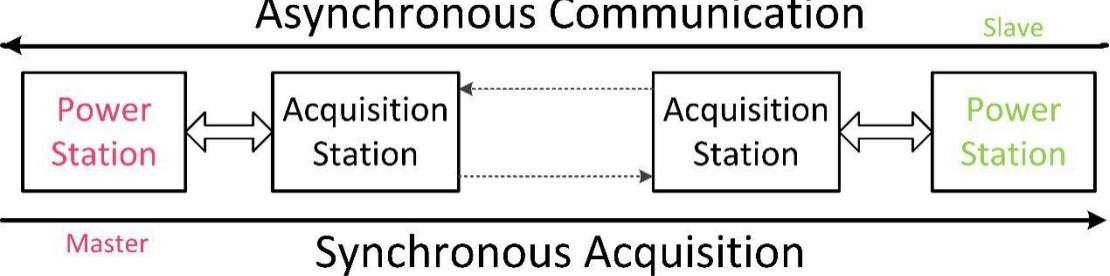

**Figure 6: Schematic diagram of synchronous data-acquisition and asynchronous communication of the data-acquisition station.**



### 3.3 High-precision synchronous data acquisition technology

As one of the important indicators to measure instrument performance, synchronization precision determines the quality of the acquired data. Synchronous data acquisition means that all acquisition stations on the acquisition chain collect data simultaneously after receiving the acquisition command. Therefore, eliminating delay in synchronous acquisition commands to reach each acquisition station is a key problem to be solved (Wang et al., 2011; Zhang et al., 2016). Delay in data frame transmission along the acquisition chain can be categorized as follows: delay due to time required on the transmission line, delay due to time spent in acquisition stations in the direction of synchronous transmission, delay due to processing time spent in slave power stations, and delay due to time spent in acquisition stations in the direction of asynchronous transmission. Figure 7 shows the delay of data frames when transmitted back and forth between the slave power station and the master power station when there are four acquisition stations in between.

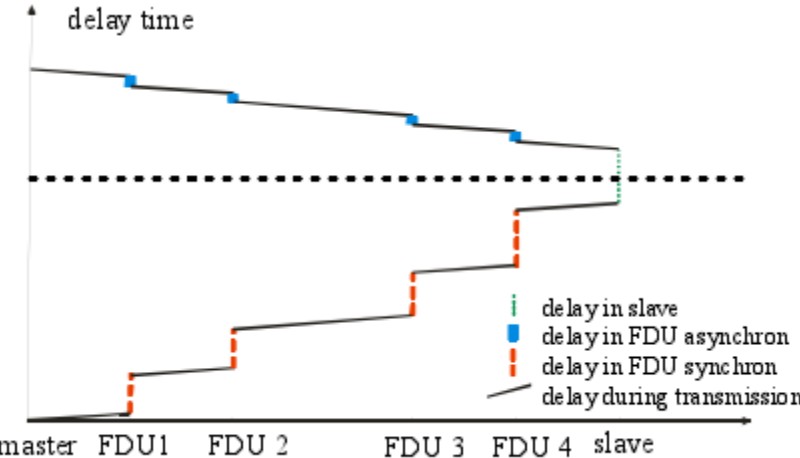

**Figure 7: Schematic of the round-trip transmission delay in data frames between two power stations**

In this study, high-precision synchronization is realized through controlling the acquisition stations by sending synchronous acquisition commands from the master and slave power stations. The master power station sends one frame of data to the acquisition stations (in synchronous acquisition mode), and the acquisition station start to decode the data frame after receiving it. If a synchronous acquisition command frame is recognized, the acquisition station forwards the data frame to the next acquisition station and starts timing simultaneously. When the same acquisition station receives the synchronous acquisition command frame again (which is returned from the slave power station in asynchronous transmission mode), it stops timing. Hence, if the timed duration in the $N^{th}$ acquisition station on the acquisition chain is $T_N$, then:

$$T_N = T_A + T_S + T_{sl},$$ (1)

$$T_d = T_S - T_A,$$ (2)

$$T_N + T_d = 2T_S + T_{sl},$$ (3)





$$T_S = \frac{1}{2}(T_N + T_d - T_{sl}) \tag{4}$$

where $T_A$, $T_S$, and $T_{sl}$ are the time durations required by the synchronous acquisition command frame in the asynchronous transmission process, in the synchronous acquisition process, and in forwarding from the slave power station, respectively. $T_D$ is the time difference between synchronous acquisition and asynchronous transmission. Since the time spent transmitting data over cables is the same in both directions for synchronous acquisition and asynchronous transmission, $T_D$ can represent the sum of the processing time differences between the two different data transmission modes in the acquisition stations. Assuming that in the $j^{th}$ acquisition station, the time consumed by the synchronous acquisition process is $t_j$ and the time consumed by the asynchronous transmission process is $t'_j$, if there are a total of $M$ acquisition stations on an acquisition chain, then:

$$T_d = \sum_{j=N+1}^{M} (t_j - t'_j) \tag{5}$$

$$T_S = \frac{1}{2}(\sum_{j=N+1}^{M} (t_j - t'_j) + T_N - T_{sl}) \tag{6}$$

During the data frame transmission process, $t_j$, $t'_j$, and $T_{sl}$ can be obtained in actual testing. Establishing the same lookup table for each acquisition station allows them to look up their corresponding delay time according to their own location information. In this way, when acquisition stations receive a synchronous acquisition command frame, each of them will delay the corresponding time duration to achieve the purpose of synchronous acquisition. Figure 8 shows the user interface of a four-channel oscilloscope, which is used to observe the synchronous signal outputs from four acquisition stations. After repeated tests and long-term observation, it was confirmed that the synchronous acquisition precision is better than 200 ns, which meets the needs of actual field exploration.

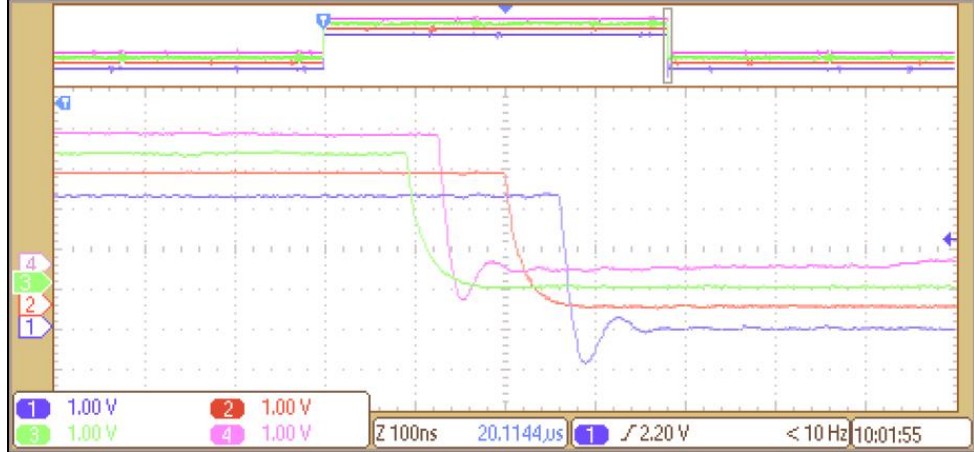

**Figure 8: Synchronization precision test results of acquisition stations.**



## 4 Summary of acquisition station test and performance

### 4.1 Equivalent input noise test

The equivalent input noise (EIN) of the acquisition station, which determines the instrument's ability to resolve weak and small signals, consisted of a variety of noises. During the test, a 1 kΩ terminal test resistor was connected to the analog-signal input

terminal, and the sampling rate of ADS1271 was configured as 1000 SPS. The data of the amplifier at different gains were collected and then processed and analyzed using MATLAB. Figure 9 shows the noise waveform diagram thus obtained.

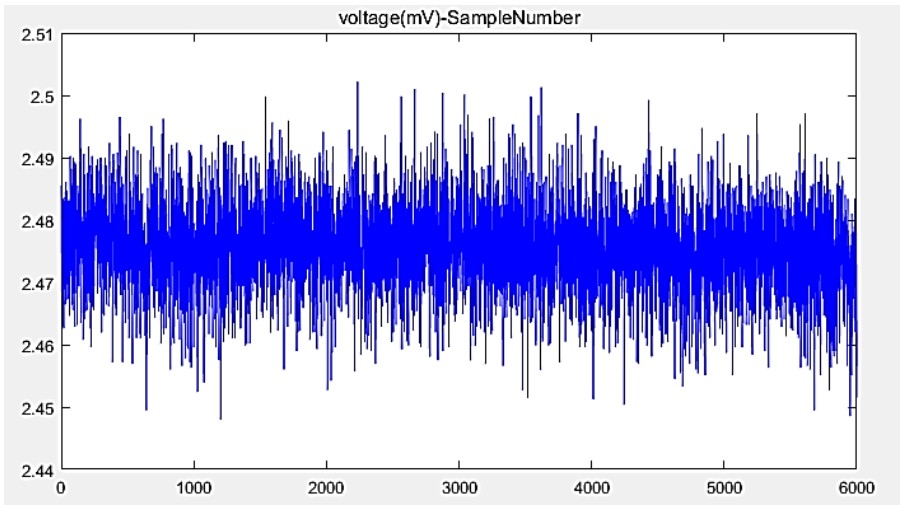

**Figure 9: EIN waveform diagram.**

The calculation results of EIN at different gains for the acquisition station are shown in Table 2.

**Table 2: Results of EIN tests.**

| Amplifier gain (dB) | EIN (µV) |
|---|---|
| 0 | 5.92 |
| 20 | 0.61 |
| 40 | 0.32 |
| 60 | 0.27 |

### 4.2 Channel crosstalk test

Since there are two channels on the analog board, they will inevitably cause mutual interference. During the test, one channel was short-circuited, and a sine wave with a frequency of 31.25 Hz and a peak-to-peak value of 4 V was input into the other channel, with the gain set to 1. Data acquisition was performed at different sampling rates, and then MATLAB was used to

obtain waveform diagrams, as shown in Fig. 10. Meanwhile, after performing fast Fourier transform (FFT) processing for the





data, it was found that at a frequency of 31.25 Hz the relative power of the input signal channel is much larger than that of the noise in the short-circuited channel, indicating that the level of crosstalk between the channels satisfies design requirements.

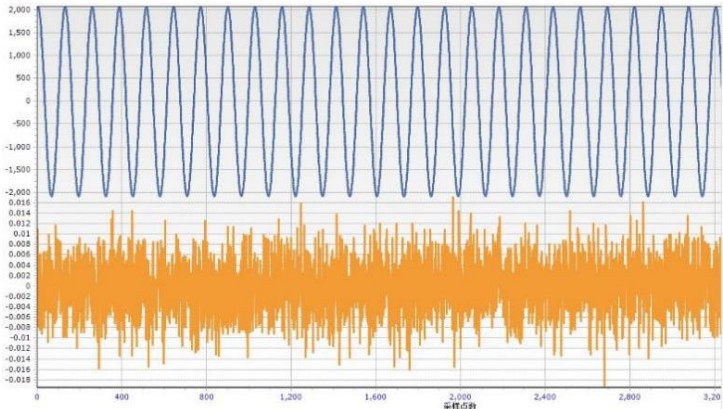

**Figure 10: Waveform of each channel during the crosstalk test.**

5  **4.3 Summary of performance indicators**

Laboratory and field test analysis show that the functions and performance indicators of the acquisition station developed in this study meet the expected design requirements. The main performance indicators are shown in Table 3.

**Table 3: Acquisition station main performance indicators.**

| | |
|---|---|
| ADC Resolution | 24 bits |
| Sampling Rates | 1,2,4, and 8 kHz |
| Preamplifier Gain | 0 dB to 60 dB in 20 dB steps |
| Synchronization Accuracy | < 200 ns |
| Maximum Input Signal | 2.5 V peak @ Gain 0 dB |
| EIN | 0.61 uV @ 1 kHz @ Gain 20 dB |
| Total Dynamic Range | 107 dB @ Gain 0 dB |
| Power Consumption | 230 mV |
| Acquisition Station Interval | 55 m (maximum) |
| Supply Voltage | 18-72 VDC |
| Channel Crosstalk | -109 dB @ Gain 0 dB |
| Common Mode Rejection Ratio | > 101 dB |
| Total Harmonic Distortion | < 0.0005% |
| Data Transmission Speed | 16 Mbps |
| Operating Temperature | -30 °C to +70 °C |


## 5 Conclusions

Based on SoPC technology, this study develops a new type of distributed seismic-electrical hybrid acquisition station. The paper mainly explores the following technical aspects:

(1) High-precision integrated seismic-electrical acquisition technology. Different front-end signal conditioning circuits were

designed based on the characteristics of seismic-electrical signals, and a 24-bit Δ-Σ ADC was used to achieve high-precision data acquisition.

(2) High-speed low-power data transmission technology. A dedicated data transmission protocol and data frame format were designed. In addition, based on the improved LVDS data transmission technology independently developed, functions of synchronous acquisition and asynchronous transmission were developed according to actual needs, to enable low-power data

transmission at a speed of 16 Mbps along a 55 m cable.

(3) High-precision synchronous acquisition technology. Acquisition stations were controlled by sending synchronous acquisition command frames from both the master and slave power stations, with a synchronization precision better than 200 ns.

(4) Highly integrated hardware circuits. In addition to signal conditioning and ADC circuits on the analog board, the digital

circuits of the acquisition stations and the key technologies, which are protected by exclusively held intellectual property rights, were integrated in a single-chip SoPC for subsequent streaming and upgrade.

*Author contributions.* The author worked as the software design and post-debugging throughout the development process, as well as the drafting of the manuscript.


*Acknowledgements.* This work is supported by the Natural Science Foundation of China (No. 41574131 and No. 41204135), the National "863" Program of China (No. 2012AA061102 and No. 2012AA09A20102), the National Major Scientific Research Equipment Research Projects of China (No. ZDYZ2012-1-05-01), the Fundamental Research Funds for the Central Universities of China (No. 2652015213), and the China Scholarship Council.

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
