# Peer review of "Development of a new distributed hybrid seismic-electrical data acquisition station based on system–on-a-programmable-chip technology"

_Geoscientific Instrumentation, Methods and Data Systems, 2019_

## Referee Comment (RC1) · Anonymous Referee #1 · 30 Apr 2019

The manuscript describes the technical realization of a new distributed data acquisition station for geophysical exploration. The digital circuits of the acquisition stations and the key technologies were integrated in a single-chip SoPC. I believe that the technical aspects explored in this study will drive the advancement of prospective integrated seismic-electrical technologies. The findings of this study are very suitable for the GI journal for Geoscientific Instruments. The manuscript is written in an adequate style and the quality of the figures is good. I have attached below several comments arising from the paper, which might be helpful to further improve the quality of the publication

of the paper. Figure 3, the DC-DC part of the power cord has too many colors. It is recommended to remove the power cord because there is no need to detail the power supply of each module in the block diagram. The format of some of the references should be properly adjusted to ensure that the article format is more standardized. A final checking for any missed spelling errors may be necessary. I am looking forward to the final publication of the manuscript.

---

## Referee Comment (RC2) · Anonymous Referee #2 · 7 May 2019

The paper is relevant to the present, describes a system that combines various methods of geophysical research. Particular attention in the paper is paid to the synchronization of measurements from a large number of stations that measure both seismic fluctuations and electric. On the scientific side, another method of synchronizing a large number of measurements without using GPS system is shown. Comments: 1. The article does not show clearly enough how to synchronize at SoPC and ADC ADS1271 levels. As we know, delta-sigma ADCs do not like frequent interruptions in data processing, which leads to a deterioration of their measurement capabilities.

[Figure]

Please, comment. 2. Figure 3 shows the structure diagram of the analogue part of the device, which shows that the seismic channel is different from the electric structure, which will result in different time delays in these channels. Also, in this case, attention should be paid to the identity of the LPF parameters both in different channels and in different stations. Please, explain this. 3. On Fig. 7 it would be good to show the time scale of the delay and the time itself. 4. In section 4, to test the noise characteristics of the station, it would be good to use the spectral characteristics that better indicate the noise of the device at different frequencies. 5. Fig. 10, it would also be nice to add the spectral characteristics of these signals. 6. Table 3 - Power consumption is measured in mW, not in mV. 7. The article states the accuracy of the synchronization should be 200 nsec, when approximate calculations show that for a frequency of 31.25 Hz (test signal in the article) and the dynamic range of this ADC, it should be at least 100 nsec. Please, clarify.

---

## Referee Comment (RC3) · Anonymous Referee #3 · 23 May 2019

This is really a good work. The authors successfully developed a new geophysical system based on the system–on-a-programmable-chip technology. In my personal view, the developed station features the following advantages: small in size, better performance, low cost and long life cycle management. Additionally, with two channels for independent seismic and electric data sampling, it allows us to make multi-parameter measurements much easier with low cost. And hence an integrated interpretation can be performed simultaneously. As a distributed system for weak geophysical data acquisition, the authors made great efforts in time synchronization and noise reduction.

To improve more in the MS and the new station, I'd like to make following suggestions:
1. Though the present values of synchronization (200ns) and noise level (0.6uV) are acceptable, further reduction are needed for higher sampling frequency and weaker signals (electric signals in geoscience usually at âľľuV or <uV). 2. To be understood easily, it is better to show the equations (1)-(2) in a graphic style. 3. Figs 7, 8, 9, 10 should be redrawn to make them clear and formal for publication. Parameter name and scale are always necessary for the vertical and horizontal axis of the figures. 4. Text and the English need much revisions (especially the INTRODUCTION). Details see the revision manuscript attached. 5. Other suggestions and comments see the revision MS attached.

Please also note the supplement to this comment:
https://www.geosci-instrum-method-data-syst-discuss.net/gi-2019-12/gi-2019-12-RC3-supplement.pdf

**Supplement:**

[revised manuscript text omitted]

---

## Author Comment (AC1) · 25 Jun 2019

Thank you very much for your affirmation of my work and suggestions for the manuscript. I have modified the Figure 3 to make it more clear.In order to avoid spelling mistakes, I checked the article again and adjusted the reference format to make it more standardized.Thanks again for your suggestions.
* * *
[Figure]

**Digital Interface Control**

**Analog Switch ADG419**

**Low-Pass Filter**

**PGA AD8253**

**AD Driver THS4521**

**ADS1271**

Detector

**Test Signal AD780**

**Analog Switch ADG419**

**Preamplifier LT1168**

**50Hz Filter**

**Low-Pass Filter**

**AD Driver THS4521**

**ADS1271**

Electrode

**Fig. 1.** Block diagram of the analog board of the data-acquisition station

---

## Author Comment (AC2) · 22 Jul 2019

(1)Higher synchronization accuracy and lower noise levels are our constant goals. And in new acquisition stations, we have used the new GPS technology to improve the synchronization accuracy and developed new analog circuits to reduce noise levels. These new results will be shown in our next papers. Thank you for your suggestions. (2) You are quite right and I will use a graph to show the equations (1)-(2). (3) I will redraw these figures to make them clear and add parameter name and scale for each figure to make them formal for publication. (4) Thank you for your revision manuscript

attached and I will carefully revise text and English of the manuscript. (5)Thank you again for your valuable suggestions and comments and I will revise the manuscript according to your suggestions.
* * *
[Figure]

synchronous acquisition
command frame from
Master Power Station

Acquisition Station N

$T_A$

Slave Power Station

$T_S$

$T_N$

$T_{sl}$

→ Synchronous Acquisition
← Asynchronous Communication

**Fig. 1.** Schematic diagram of the delay time of the Nth acquisition station

delay time

delay in SPS

delay in AS asynchron

delay in AS synchron

delay during transmission

station type

MPS    AS1    AS2    AS3  AS4   SPS

**Fig. 2.** Schematic of the round-trip transmission delay in data frames between two power stations(MPS,Master Power Station;AS,Acquisition Station;SPS,Slave Power Station)

[Figure]

Fig. 3. Distribution of the EIN of a data acquisition station

---

## Author Comment (AC3) · 25 Jul 2019

Thank you very much for your affirmation of my work and suggestions for the manuscript. (1) The clock of the ADC is divided by the high-precision clock source of SoPC. SoPC controls the ADS1271 through the SPI protocol, and uses the SYNC signal of the ADS1271 to realize synchronous data acquisition of each channel. At the same time, we all know that the data of the delta-sigma ADC just started to collect is unreal. We have performed corresponding operations during data processing to ensure the authenticity and validity of the data. SoPC uses FIFO to buffer data,

avoiding the frequent interruption of ADC work. I wish my explanation can help you. I have added the relevant content in section 2.3. (2) As you said, the seismic channel is different from the electric structure. We have different commands to control the data acquisition of the two channels. The main control board can identify the channel type, calculate different delay time, and achieve high-precision synchronous acquisition of two channels according to the time. As for the LPF parameters, the cutoff frequency of the seismic channel is about 1.2 kHz and the cutoff frequency of the electrical channel is approximately 3.4 kHz. I have added the differences between the LPF parameters in the seismic channel and the electrical channel in Section 3.1. (3) I have redrawn Figure 7 and added the time scale of the delay. But please note that Figure 7 is only a schematic diagram to help the reader understand the transmission delay. The specific time needs to be calculated according to the formula and the actual test. (4) Thanks for your suggestion. I will add the spectral characteristics. (5) I will add the spectral characteristics of these signals. (6) I have corrected this error. (7) Sorry, I don't understand the question very well. The 200ns mentioned in the manuscript is the time synchronization accuracy between the various acquisition stations and it is related to delay due to time required on the transmission line, delay due to time spent in acquisition stations in the direction of synchronous transmission, delay due to processing time spent in slave power stations, and delay due to time spent in acquisition stations in the direction of asynchronous transmission. It has nothing to do with the frequency of the acquired signal and the dynamic range of the ADC. Thank you again for your comments.

---

## Author Comment (AC4) · 25 Jul 2019

Thank you very much for your affirmation of my work and revision manuscript attached.

I have tried my best to revise the manuscript based on each comment in the manuscript attached. I rewrote the INTRODUCTION and modified many sentences and vocabulary according to your suggestions. What's more, I have redrawn some figures to make them formal for publication.

Thank you again for your revision manuscript attached.